# Understanding the Biophysical Interaction of LTX-315 with Tumoral Model Membranes

**DOI:** 10.3390/ijms24010581

**Published:** 2022-12-29

**Authors:** Maria C. Klaiss-Luna, Małgorzata Jemioła-Rzemińska, Kazimierz Strzałka, Marcela Manrique-Moreno

**Affiliations:** 1Chemistry Institute, Faculty of Exact and Natural Sciences, University of Antioquia, A.A 1226, Medellin 050010, Colombia; 2Faculty of Biochemistry, Biophysics and Biotechnology, Jagiellonian University, 30-392 Krakow, Poland; 3Malopolska Centre of Biotechnology, Jagiellonian University, Gronostajowa 7a, 30-387 Krakow, Poland

**Keywords:** LTX-315, anticancer peptides, lipid bilayer, model membranes, peptide–membrane interaction

## Abstract

Host defense peptides are found primarily as natural antimicrobial agents among all lifeforms. These peptides and their synthetic derivatives have been extensively studied for their potential use as therapeutic agents. The most accepted mechanism of action of these peptides is related to a nonspecific mechanism associated with their interaction with the negatively charged groups present in membranes, inducing bilayer destabilization and cell death through several routes. Among the most recently reported peptides, LTX-315 has emerged as an important oncolytic peptide that is currently in several clinical trials against different cancer types. However, there is a lack of biophysical studies regarding LTX-315 and its interaction with membranes. This research focuses primarily on the understanding of the molecular bases of LTX-315′s interaction with eukaryotic lipids, based on two artificial systems representative of non-tumoral and tumoral membranes. Additionally, the interaction with individual lipids was studied by differential scanning calorimetry and Fourier-transformed infrared spectroscopy. The results showed a strong interaction of LTX-315 with the negatively charged phosphatidylserine. The results are important for understanding and facilitating the design and development of improved peptides with anticancer activity.

## 1. Introduction

Host defense peptides (HDPs) are essential components of the chemical armory found in nature as part of the innate immune systems of all living organisms [1]. These molecules are mainly cationic at physiological pH, amphipathic, and relatively small (up to 50 amino acids), making them complex and diverse in function [2,3,4]. HDPs have been recognized primarily as antimicrobial agents. However, in the last decade they have shown a broad spectrum that includes anti-biofilm, antifungal, antiviral, and antitumor activity through a membrane-related mechanism of action [5,6,7,8,9,10]. For all of these reasons, they have emerged as promising candidates for new approaches in the treatment of various pathologies [1,11].

One of the most attractive properties of peptides is that natural sequences can be chemically modified and optimized to potentiate their activity. These modifications are based on the variation of physicochemical properties such as charge, hydrophobicity, and including substitutions for non-conventional amino acids. Among the most recently reported synthetic peptides, LTX-315 is a promising 9-mer synthetic peptide designed and modified from the host defense peptide lactoferricin, with a net positive charge of +6 at physiological pH [12,13]. Its activity against advanced melanoma, soft-tissue sarcoma, basal-cell carcinoma, and several human and animal models (such as melanocytes, fibroblasts, and endothelial cells) has been reported [14,15,16,17]. These results showed values of IC_50_ between 12.3 and 27.6 µM. The experimental preclinical studies of LTX-315 in animals showed that intratumoral application of 50 µL of the peptide at 20 mg/mL induced growth inhibition, complete regression, controlled metastatic dissemination, and activation of the immune response [13,16,18].

LTX-315 is currently considered the first-in-class oncolytic peptide evaluated in human clinical trials to treat several types of cancer (https://clinicaltrials.gov (accessed on 20 October 2022) Identifier: NCT04796194, NCT03725605, NCT05188729) [12,19]. Like most HDPs, the membranolytic activity of LTX-315 against tumoral cells has been explained by a destabilizing mechanism of the lipid bilayer. The most accepted mechanism is mediated in the first step by electrostatic interactions [20], based on differences in composition between non-tumoral and tumoral cell membranes [21]. Lipids are asymmetrically distributed in cell membranes. In a eukaryotic membrane, zwitterionic phospholipids such as phosphatidylcholine (PC) and sphingomyelin (SM) are in a higher proportion in the exoplasmic leaflet, which confers an overall neutral charge, while phosphatidylethanolamine (PE) and negatively charged phosphatidylserine (PS) are located mainly in the cytosolic leaflet [22,23,24]. However, in the case of tumoral cells, it has been reported that the asymmetric distribution of phospholipids between the two leaflets of the cell membrane is lost by altering homeostatic mechanisms such as cancer [24,25]. This causes PS overexpression in the exoplasmic leaflet, contributing to the characteristic negative charge of tumoral cells, as well as the presence of heparin sulfate side chains from proteoglycans and *O*-glycosylated mucins [26,27]. Therefore, the surface of the cancer cell membrane acquires a negative charge due to the abnormal concentration of PS. The presence of PS in tumoral cell membranes favors electrostatic attraction between the negatively charged groups of the lipid and peptides, and this interaction induces instability, as well as structural and physicochemical changes in the lipid bilayer, leading to cell death through different steps [28,29]. For this reason, it has been proposed that the selectivity of LTX-315 toward cancer cells is conferred due to its cationic character, which promotes the binding between the positively charged residues of the peptide and the negatively charged groups of the tumoral cell membranes [30,31,32].

Despite the potential of LTX-315 as a pharmaceutical agent and the fact that the peptide is already in several clinical trials [19], there are not enough studies establishing the molecular bases of the interaction of the peptide with membranes. For this reason, this study aimed to evaluate the interaction between LTX-315 and two multicomponent lipid systems representative of the tumoral and non-tumoral cell membranes by differential scanning calorimetry (DSC) and Fourier-transform infrared (FTIR) spectroscopy. [33]. However, to understand whether the peptide selectively interacts with one of the lipids of the multicomponent systems, we also evaluated the effects of LTX-315 on pure PC, PE, SM, and PS liposomes. In addition, FTIR studies were performed to evaluate the secondary structure of LTX-315 in the presence of tumoral and non-tumoral model membranes. The conformational change in the structure of a membrane-active peptide has been proposed as a fundamental step of the activity of host defense peptides [34,35].

## 2. Results

### 2.1. Prediction of LTX-315 Structure

LTX-315 is a first-in-class synthetic oncolytic peptide with a +6 charge under physiological conditions. This peptide was designed and chemically modified through structure–activity studies by Lytix Biopharma AS [13,36]. The chemical structure is represented in Figure 1a. The structure contains three tryptophans and one non-conventional amino acid—diphenylalanine, which contributes a 40% hydrophobicity. The helical wheel projection of the peptide calculated using EMBOSS is represented in Figure 1b. It shows an amphipathic structure with two faces: one conserving the hydrophobic residues, and the other forming a polar surface through the presence of the lysines. Most HDPs have been extensively reported to frequently unfold in aqueous environments but fold during interactions with membranes [37]. The prediction of the secondary structure of LTX-315 obtained by PEP-FOLD3 revealed a random structure at physiological pH (Figure 1c). Among the user-friendly computational tools, PEP-FOLD3 has recently become an excellent alternative method to study peptides in solution, as it allows the generation of native and near-native models represented with the best clusters [38].

### 2.2. Differential Scanning Calorimetry (DSC) of Tumoral and Non-Tumoral Model Membranes

Calorimetric experiments were performed in order to determine the effects of LTX-315 on the thermotropic properties of two multicomponent systems representative of non-tumoral and tumoral membranes. The results of the incubation of LTX-315 with the non-tumoral model membrane are summarized in Figure 2a. The thermogram of the non-tumoral model membrane in the absence of LTX-315 showed a broad endothermic peak occurring at 40.1 °C, with a corresponding phase transition enthalpy ΔH = 26.63 kJ mol^−1^. This transition corresponds to the conversion of the gel phase (L_β_) to the lamellar liquid-crystal (L_α_) phase of the mixture that does not contain PS. At concentrations of 1 and 5 mol% LTX-315, there were no significant changes in the thermal behavior of the non-tumoral membranes. At a higher concentration of 10 mol%, the presence of an additional peak was evident at approximately 44.5 °C.

The heating thermograms of the liposomes for the representative tumoral cell systems are summarized in Figure 2b. The control experiment in the absence of LTX-315 showed a broad endothermic peak, very similar to the non-tumoral membrane, with a transition at 40.5 °C and a ΔH = 33.34 kJ mol^−1^. At the lowest evaluated concentration of 1 mol%, the peptide induced an important reduction in the peak height, reflecting how the interaction of the peptide affects the thermotropic event. At higher concentrations of LTX-315 (i.e., 5 and 10 mol%), the effect was stronger than the observed for the non-tumoral liposomes. The thermograms exhibit the concentration-dependent effect of the peptide and a mild shift of the T_m_ value to lower temperatures. At the concentration of 10 mol%, apart from a further decrease in the height of the main transition peak, the peptide induced the appearance of a pronounced shoulder at 44.5 °C. The results suggest a destabilization effect of the peptide, the ability to alter the phase transition of the tumoral system, and a stronger induction of phase separation in comparison with that of the non-tumoral system.

Since the thermograms presented in Figure 2 correspond to multicomponent lipid systems, a deconvolution procedure was performed to check whether the components of the mixture were affected differently by the presence of LTX-315 (Figure 3). As a result of deconvolution, three peaks were found to fit the calorimetric profiles best. The lowest temperature component was characterized by the highest value of full width at half-maximum (∆T_1/2_) and the lowest height of the peak. An apparent difference was found between non-tumoral and tumoral systems regarding the share of peaks #2 and 3. Centered at about 39 and 42 °C, respectively, they contributed much more evenly in the case of endotherms registered for PS-depleted membranes than that for tumoral counterparts. At the lowest concentration of LTX-315 (1 mol%), there was a slight change in the ΔH peaks #1 and #2 in the thermograms of both the non-tumoral and tumoral lipid systems, without any significant changes in temperatures. However, at the concentration of 5 mol%, the differences in the interaction between the peptide and lipids were stronger in the two investigated lipid systems. Upon deconvolution, the heating endotherm of the tumoral model membrane revealed a significant change in the contributing components in favor of the lower temperature peak #2. This tendency was also kept for the highest investigated concentration of peptide (10 mol%); however, in this case, a distinct shoulder was clearly seen, associated with the increasing height of peak #3. Interestingly, deconvolution of non-tumoral systems showed that the lower temperature component (peak #2) became dominant only at 10 mol% peptide content. The thermodynamic parameters of the phase transition of both lipid systems in the presence of LTX-315 are presented in Table 1.

To understand the potential selective interaction of LTX-315 with one or more of the main lipids in eukaryotic membranes, pure liposomes of DPPC, SM, DPPE, and DPPS were prepared and incubated with different concentrations of peptide. The results of the DPPC heating thermograms are represented in Figure 4a. The thermogram of the pure DPPC shows two transitions: The small peak at the lower temperature (36 °C), known as pretransition, corresponds to the transition from the gel phase (L_β′_) into the ripple phase (P_β′_), with ΔH = 32.4 kJ mol^−1^. The higher peak, at 42 °C, is the main transition from P_β′_ to the liquid-crystalline phase (L_α_), with ΔH = 2.8 kJ mol^−1^. These data are consistent with previous reports by Çetinel and co-workers [38]. Incubation of DPPC liposomes with increasing amounts of LTX-315 did not show a significant effect on the thermotropic behavior of the lipid bilayers. Even at the highest LTX-315 concentration evaluated, the metastable ripple phase was present in the thermograms. The thermodynamic parameters of the pretransition and main-phase transition of DPPC are presented in the Appendix A. The heating thermograms of DPPE liposomes containing LTX-315 are presented in Figure 4c. DPPE in the well-hydrated state presents a single, strong, and acute transition peak at 65 °C with a transition enthalpy of 35.6 kJ mol^−1^ (see Appendix A), which corresponds to the conversion from the gel phase to the liquid-crystal phase. This result is very similar to the value of 34.7 kJ mol^−1^ reported in previous reports by Lewis and collaborators [39]. Different concentrations of LTX-315 did not induce a significant effect on the thermotropic behavior of the DPPE liposomes.

The results of the heating thermograms of the fully hydrated SM liposomes with different concentrations of LTX-315 are represented in Figure 4b. The endotherm representing the gel–liquid-crystalline phase transition for pure sphingomyelin showed a phase transition temperature of 39.7 °C and ΔH = 40.2 kJ mol^−1^. The pretransition was not observed in this case, and the peak was broader in comparison with the other phospholipids, which could be explained by the heterogeneous acyl chain composition of egg sphingomyelin. The analysis of the results showed that the peptide significantly affected the thermotropic properties of the SM vesicles in a concentration-dependent manner. Increasing concentrations of LTX-315 induced a broadening of the endothermic peak. At 5 and 10 mol% LTX-315, the peptide induced phase separation, and the presence of a shoulder was evident at about 42.5 °C. The thermodynamic parameters of the phase transition of SM are presented in the Appendix A. Finally, the heating thermograms of the DPPS liposomes in the presence of different concentrations of LTX-315 are summarized in Figure 4d. The heating curve of the pure DPPS liposomes in the absence of LTX-315 is characterized by an endothermic peak with the main transition at 54 °C and ΔH = 35.5 kJ mol^−1^. A temperature of 53.4 °C and ΔH = 36 kJ mol^−1^ were reported by Çetinel and collaborators [38]. Increasing concentrations of LTX-315 induced a gradual phase separation in the DPPS liposomes. At a concentration of 5 mol%, an additional peak was present at about 54.7 °C. At the highest concentration of LTX-315 evaluated (10 mol%), the latter peak was flattened, but one more appeared at about 51 °C. The thermodynamic parameters of the phase transition of DPPS are presented in the Appendix A.

### 2.3. Phase Transition Measurements

The thermotropic behavior of membrane lipids can be observed by FTIR from the changes in the symmetric stretching band of the methylene groups (ν_s_CH_2_) from the acyl chains of the lipid bilayer. Figure 5 shows the temperature dependence of the wavenumber values of the peak positions of both multicomponent lipid systems. The non-tumoral system and the tumoral system in the absence of LTX-315 had a T_m_ of 39.7 and 40.4 °C, respectively. The incubation of different concentrations of LTX-315 with the non-tumoral system showed that the peptide induced an increase in the wavenumbers of the gel phase and the liquid-crystalline phase (Figure 5a). This effect could be observed when the wavenumber position was analyzed at fixed temperatures below and above the T_m_. With increasing concentrations of LTX-315, there was a shift in T_m_ to lower temperatures. However, in the non-tumoral system, above the concentration of 2.5 mol% LTX-315 there was no significant effect of other, higher concentrations. At the highest concentration evaluated, T_m_ was decreased by 1.4 °C (See Table 2).

The results of LTX-315 incubation with the tumoral membrane are summarized in Figure 5b. The results showed that at the same concentrations evaluated, LTX-315 induced significant effects on lipid packing in the gel and liquid-crystalline phases. At fixed temperatures below and above the T_m_, the increases in the wavenumbers were higher in comparison with the non-tumoral system. The increasing concentrations of the peptide also shifted the T_m_ to lower temperatures, in a concentration-dependent manner. At the highest concentration evaluated (10 mol%), the change in T_m_ was 1.6 °C (see Table 2). These results showed a stronger effect of LTX-315 on the tumoral membrane.

The asymmetric stretching vibration of the PO_2_^−^ moiety has been extensively studied as a sensor of interactions in the headgroup region of lipids [40,41,42]. The ν_as_PO_2_^−^ band is very sensitive to the hydration conditions; in fact, the wavenumber values decrease substantially with increasing hydration, where the band is shifted from 1250 cm^−1^ in a dry state or poorly hydrated conditions to 1220–1227 cm^−1^ for highly hydrated bilayers. The change in the wavenumbers is caused by the hydrogen bonding of water molecules to the charged phosphate group. Analysis of the ν_as_PO_2_^−^ vibration region of the non-tumoral and tumoral systems showed that the ν_as_PO_2_^−^ vibration was present for both systems at 1223 cm^−1^, indicating that the systems were highly hydrated. However, increasing concentrations of LTX-315 did not affect the wavenumber position of the ν_as_PO_2_^−^ vibration.

A deeper analysis of the interfacial phospholipid region was performed to study whether increasing concentrations of LTX-315 induced changes in the hydration conditions of the non-tumoral and tumoral model membranes. The carbonyl group of the phospholipid is another sensitive sensor for the hydration of the lipids, but in the interface region and sensitive to the polarity, degree, and nature of the hydrogen bonding interactions. Different hydrations of the carbonyl groups lead to splitting of the group vibration into two different bands. The deconvolution of the band is composed of two bands: 1723.9 cm^−1^ for the hydrogen-bonded C=O group, and 1736.5 cm^−1^ for the non-hydrogen-bonded C=O group [43,44]. The analysis of the results showed that in the liquid-crystalline state, increasing the concentration of LTX-315 induced a shift of approximately 2 cm^−1^ in the C=O bands of the tumoral system to higher wavenumbers.

### 2.4. Determination of the Secondary Structure of LTX-315

According to the literature, most peptides undergo a conformational change when they bind to cell membranes. This conformational change has been proposed as an essential step of the activity of HDPs. For this reason, the determination of the secondary structure of LTX-315 in buffer and the conformational change when the peptides were incubated with the two representative multicomponent lipid systems was evaluated. The results of the secondary structure analysis showed that LTX-315 presented a random coil structure in buffer and in the presence of both multicomponent lipid systems. The results highlight that LTX-315 did not undergo a conformational change between the aqueous phase and the lipid environment.

## 3. Discussion

The increasing number of cancer cases worldwide has caused the need for new molecules to be used for new treatments or in combination with current chemotherapeutics. Among these agents, HDPs have emerged as promising candidates in cancer therapy, due to their high potential to be selective against cancer cells and exert their activity through a mechanism of action extremely hard to counteract by the cells [45]. One of the most promising candidates developed in recent years is the synthetic peptide LTX-315—also known by its trade name Oncopore™ [18,36]—which is active in several cancer cell lines and is in phase II clinical trials. LTX-315, after intratumoral injection, has been reported to alter the cell membrane’s integrity through the necrosis mode of action to induce an immunogenic response that releases danger-associated molecular patterns and tumor-associated antigens into the tumor microenvironment, resulting in cell death [14,46]. Despite the enormous potential of LTX-315, there is a lack of information on its complete mechanism of action. Since the first step of the reported mechanism involves cell membrane integrity, the study of membrane–peptide interaction becomes an important subject to fully understand the potential of LTX-315 as a pharmaceutical agent. For this reason, this study focused on studying the interactions between LTX-315 and synthetic lipid systems representative of non-tumoral and tumoral membranes, through spectroscopic and thermodynamic techniques.

The first approach was to evaluate the secondary structure of LTX-315. The secondary structure depends mainly on specific physicochemical properties such as amino acid sequence, charge, amphipathicity, and hydrophobicity [31]. Bioactive peptides have commonly been reported to unfold in solution and fold in a lipid environment as α-helices or β-sheets [26,47]. The analysis of the helical wheel projection of the peptide shows a potential amphipathic structure. In principle, what is expected is that this region might be involved in the strong initial electrostatic interactions between the peptide and the negatively charged groups present in the predominant phospholipid constituents of the tumoral membranes, such as PS. Few secondary structure predictions have been reported in the literature on LTX-315. In 2014, Camilio and co-workers predicted a helical coil structure using the Garnier–Osguthorpe–Robson V method [18], and two years later Haug et al. illustrated an idealized helical structure using PyMOL 1.3 [12]. We predicted the secondary structure of LTX-315 in solution and at neutral pH [48]. The results suggested that the peptide had a random coil structure under the experimental conditions, and this is in accordance with the literature for bioactive peptides in aqueous solutions.

Following the need to study the structure of the peptide, the conformation of LTX-315 was evaluated in buffer solution and in two different lipid environments by infrared spectroscopy. The results showed that LTX-315 has a random conformation in solution and does not change when binding or interacting with either non-tumoral or tumoral membranes. Our experimental findings by infrared spectroscopy were in accordance with the prediction of PEP-FOLD3 and the reported results of circular dichroism (CD) experiments published by Koo et al. [49]. The authors demonstrated that LTX-315 is structureless in aqueous environments, 50% ***v***/***v*** TFE and lipid vesicles prepared from 100% DOPC, and 70/30 mol% DOPC/DOPS [49]. Several bioactive peptides with anticancer activity have been reported to have an unordered conformation [50]. This is the case for PR-39—a proline-rich peptide that has shown activity against hepatocellular carcinoma cell lines and was found random conformation by CD spectroscopy in sodium phosphate buffer containing 0–20% hexafluoro-2-propanol (HFIP) [31,51]. Alloferon—an immunomodulatory peptide active against prostate, colorectal, and pancreatic cancers—is used as an adjuvant of chemotherapeutic drugs. The peptide was studied by CD spectroscopy, and the results showed the absence of a secondary structure [52,53]. It is important to understand that differences between computational and experimental methods for predicting the secondary structure may occur. While the computational tools are static, experimental systems such as FTIR are very dynamic environments, with the possibility of studying the peptide structure at a physiological temperature of 37 °C.

The next step was to evaluate the effects of LTX-315 on the thermotropic behavior of the two synthetic lipid systems. The thermograms of the non-tumoral and tumoral lipid systems were obtained by DSC. The phase transition temperatures of pure saturated lipid systems are highly cooperative events. However, the thermotropic events of multicomponent lipid systems are broader and less cooperative, as has been reported by Lewis et al. [54]. For multicomponent lipid systems, the T_m_ will correspond to a value depending on the participating lipids and their proportions. As the tumoral lipid system contained PS, the T_m_ of the model had a higher value than the non-tumoral system. Analysis of the DSC and FTIR results showed that LTX-315 induced greater destabilization of the tumoral lipid system compared to the non-tumoral system. The difference between the two systems was the presence of PS in the tumoral system, which incorporates a net negative charge on the liposome surface. However, neither the electrostatic attraction nor the hydrophobic environment of the two lipid systems induced any changes in the secondary structure of the peptide. The folding of the peptide in solution could not be favored due to the length of the amino acid sequence, in which adjacent bulky amino acids such as tryptophan and diphenylalanine created steric hindrance, and the lysine content conferred a net charge of +6 that triggered electrostatic repulsion in the structure. The results obtained by DSC and FTIR suggest that even LTX-315 with no conformational change can interact with the surface of the lipid bilayer; once there, the peptide anchors its hydrophobic residues to the membrane core. Chan et al. have described similar approaches to the membranolytic mechanism for linear peptides that are rich in tryptophan, as in the case of indolicidin [55], while Gong et al. described no conformational change of peptides during their interaction with cells [34]. LTX-315 could be anchored but not inserted into the lipid bilayer of synthetic lipid systems through tryptophan and diphenylalanine. This could be explained by the hydrophobicity and size of the peptide. The presence of the non-polar amino acids imparts a hydrophobicity of 40% in the structure, allowing these residues to interact with the hydrophobic core of the membrane, while the positively charged lysines interact with the lipids’ polar headgroups. This could explain why the peptide influences the lipid packing of the tumoral membrane more significantly than the non-tumoral model membrane. These results are supported by the analysis of the vibration bands of the polar and interfacial regions of the lipid surface. The asymmetric vibration of PO_2_^−^ was not affected by increasing concentrations of the peptide, but the C=O group in the interfacial region was affected by the interaction with LTX-315. The interaction with the cationic peptide caused its hydrophobic residues to be anchored in the lipid bilayer’s interphase, displacing water molecules and letting the methylene groups acquire higher degrees of freedom upon heating, while causing the T_m_ to shift to lower temperatures.

To understand the preferential interaction of LTX-315 with some of the lipid components of eukaryotic membranes, multilamellar vesicles were prepared with the individual lipids involved in the model lipid membranes, with different concentrations of LTX-315. According to the DSC results, there were no remarkable effects on the thermodynamic parameters of the DPPC and DPPE vesicles in the presence of the peptide in the evaluated concentration range. Conversely, SM vesicles were affected by the presence of the peptide. The results showed a decrease in enthalpy, a slight decrease in T_m_, and peak broadening until the transition was almost diminished. The analysis of the results showed that LTX-315 could be located in the aqueous/lipid interphase and was capable of significantly affecting the structure of SM vesicles. It has been reported that SM can produce several intra- and intermolecular hydrogen bonds due to the sphingosine moiety in its structure, which contributes substantially to structural changes in the polar headgroup and interfacial regions [56]. Finally, the interaction of LTX-315 with the DPPS vesicles showed the strongest destabilization effect. The interaction of the peptide induced changes in the microheterogeneity by inducing phase separation as a concentration-dependent effect, along with changes in the hydration of headgroups that caused the T_m_ to shift towards lower temperatures. Taking into account the above and the broadening of the peak, the interaction could be summarized as a favorable electrostatic interaction that induced the destabilization of the PS vesicles by the peptide molecules, possibly because of the strongest electrostatic attraction and anchoring on the lipid bilayer. The reduction in the ∆H of the transition supports the affinity for electrostatic interactions between phosphatidylserine and LTX-315, highlighting the importance of the cationic character of peptides to exert their activity more selectively toward tumoral membranes. The strong interaction with PS could also explain the stronger activity of LTX-315 against the synthetic tumoral system. This interaction could also be responsible for the phase separation in the multicomponent lipid systems.

## 4. Materials and Methods

### 4.1. Reagents

1,2-Dipalmitoyl-sn-glycero-3-phosphocholine (DPPC, Lot. 160PC-318), sphingomyelin egg chicken (SM, Lot. 860061P-25MG-A-116), 1,2-dipalmitoyl-sn-glycero-3-phosphoethanolamine (DPPE, Lot. 160PE-106), 1,2-dipalmitoyl-sn-glycero-3-phospho-L-serine sodium salt (DPPS, Lot. 840037P-500MG-A-078CL750332P-200MG-A-030), 1-palmitoyl-2-oleoyl-glycero-3-phosphocholine (POPC, Lot. 850457P-500MG-A-211), 1-palmitoyl-2-oleoyl-sn-glycero-3-phosphoethanolamine (POPE, Lot. 850757P-500MG-B-151), and 1-palmitoyl-2-oleoyl-sn-glycero-3-phospho-L-serine sodium salt (POPS, Lot. 840034P-25MG-A-250) were purchased from Avanti Polar Lipids (Alabaster, AL, USA). HEPES was purchased from Sigma-Aldrich (St. Louis, MO, USA), NaCl from Carlo Erba (Val-de-Reuil, Normandy, France), and EDTA from Amresco (Solon, OH, USA).

LTX-315 peptide (KKWWKKWDipK-NH_2_) Dip: diphenylalanine (Lot. U037QFC180-14/PE6102) was synthesized according to the sequence by solid-phase methods and purchased from GenScript (Piscataway Township, NJ, USA). The purity of the peptide was determined to be higher than 95% by analytical HPLC, TFA removal was performed, and the molecular weight was confirmed with MALDI-TOF mass spectrometry (Appendix A).

### 4.2. Prediction of LTX-315 Structure

The helical wheel projection of LTX-315 was calculated using the helical wheel predictor by EMBOSS (https://www.bioinformatics.nl/cgi-bin/emboss/pepwheel (accessed on 15 October 2022). The 3D structure prediction of the LTX-315 peptide was generated using the online Peptide Structure Prediction Server PEP-FOLD3. Since none of these computational tools support unconventional amino acids, the analyzed peptide structure included only the first 7 residues (KKWWKKW). The model was generated through the identification of the best conformational cluster from several simulations of its amino acid sequence. The structure was predicted by the hidden Markov model structural alphabet (SA-HMM) algorithm and coarse-grained force field computational modeling (https://bioserv.rpbs.univ-paris-diderot.fr/services/PEP-FOLD3/#overview (accessed on 7 October 2022) [48,57,58].

### 4.3. Differential Scanning Calorimetry (DSC) of Tumoral and Non-Tumoral Model Membranes

The lipid systems DPPC/SM/DPPE 4.35:4.35:1 (*w*/*w*) and DPPC/SM/DPPE/DPPS 4.35:4.35:1:0.3 (*w*/*w*) were evaluated to represent the non-tumoral and tumoral cell membranes, respectively [59,60]. For the calorimetric experiments, appropriate amounts of DPPC and SM were weighed and dissolved in chloroform to prepare the stock solutions. In the case of DPPE and DPPS, the lipids were weighed and dissolved in chloroform:methanol (70:30). An appropriate volume from each stock solution was used to obtain a final phospholipid concentration of 1 mM to form a thin film on the walls of a glass test tube. The solvent was dried under a stream of nitrogen, and the traces were removed by keeping the samples under reduced pressure (about 13.3 Pa) for 30 min. Dried lipids were hydrated with buffer (10 mM HEPES, 500 mM NaCl, 1 mM EDTA, pH 7.4). MLVs were formed by vortexing the samples for 6 min, followed by 1 min of sonication above the main phase transition temperature of the lipids. LTX-315 stock solution was prepared in the same buffer and added to the films, and MLVs were formed under the same conditions explained above.

DSC measurements were performed using a Nano DSC device (TA Instruments, New Castle, DE, USA). The sample cell was filled with 400 µL of MLV suspension; a reference of an equal volume of buffer was used. Cells were sealed and equilibrated for about 10 min at the starting temperature. The heating/cooling rates were 1 °C per minute, and the scans were recorded within a range of ~40 °C, defined by the lipid system. Heating scans were carried out first. The reference scan was subtracted from the sample scan. Each dataset was analyzed, and the values of transition temperatures, enthalpy, and entropy were calculated using the NanoAnalyze software package supplied by TA Instruments. At least three independently prepared samples were measured to check the reproducibility of the DSC experiments. The accuracy was ±0.1 °C for the main phase transition temperature and ±1 kJ mol^−1^ for the main phase transition enthalpy.

### 4.4. Infrared Spectroscopy Experiments

Supported lipid bilayers (SLBs) were prepared in situ in a BioATR II cell. The unit was integrated with a Tensor II spectrometer (Bruker Optics, Ettlingen, Germany) with a liquid nitrogen MCT detector using a spectral resolution of 4 cm^−1^ and 120 scans per spectrum. The desired temperature was set by a Huber Ministat 125 computer-controlled circulating water bath (Huber, Offenburg, Germany) with an accuracy of ±0.1 °C. First, the background was taken using 10 mM HEPES buffer, 500 mM NaCl, and 1 mM EDTA in the temperature range of 30–50 °C. Subsequently, to coat the silicon crystal, stock solutions of the different lipid systems were dissolved in chloroform. The preparation of the stock solutions was carried out depending on the lipid system to be analyzed. For non-tumoral and tumoral membrane composition, this was DPPC/SM/DPPE 4.35:4.35:1 (*w*/*w*) and DPPC/SM/DPPE/DPPS/4.35:4.35:1:0.3 (*w*/*w*), respectively [59,60]. The cell was filled with 20 µL of the lipid stock solution, and the solvent was evaporated, resulting in a lipid multilayer film. For in situ measurements, the cell was subsequently filled with 20 µL of buffer or peptide solution and incubated over the phase transition temperature for 15 min.

For the evaluation of the gel–liquid-crystalline phase behavior, the peak position of the symmetric stretching vibration of the methylene band ν_s_ (CH_2_) around 2970–2820 cm^−1^ was taken, which is a sensitive marker of lipid order. Furthermore, vibrational bands from the interface region (ester carbonyl stretching around 1725 to 1740 cm^−1^) and the headgroup (phosphate antisymmetric stretching at 1280 to 1130 cm^−1^) were analyzed. To determine the position of the vibrational bands in the range of the second derivatives of the spectra, all of the absorbance spectra in this range were cut and shifted to a zero baseline, and the peak picking function included in OPUS software was used. In each case, the results were plotted as a function of the temperature. To determine the transition temperature (T_m_) of the lipids, the curve was fitted according to the Boltzmann model to calculate the inflection point of the obtained thermal transition curves using the OriginPro 8.0 software (OriginLab Corporation, USA). In the case of overlapping bands—especially for the analysis of the ester carbonyl and phosphate stretching vibrations—curve fitting was applied. An estimation of the number of band components was obtained from the second derivatives and deconvolution function. The band shapes of the single components were superpositions of Gaussian and Lorentzian bands. The instrumental wavenumber resolution was better than 0.02 cm^−1^, and the wavenumber reproducibility in repeated scans was better than 0.1 cm^−1^.

### 4.5. Determination of the Secondary Structure of LTX-315

LTX-315 peptide solution was prepared at a 2 mM concentration in buffer (10 mM HEPES, 500 mM NaCl, 1 mM EDTA, pH 7.4). Appropriate amounts of POPC/SM/POPE 4.35:4.35:1 (*w*/*w*) and POPC/SM/POPE/POPS 4.35:4.35:1:0.3 (*w*/*w*) for the non-tumoral and tumoral membranes [59], respectively, were weighed in order to obtain a 6 mM final concentration of the representative liposomes. The lipids were dissolved in pure chloroform in a glass test tube, the solvent was dried under a stream of nitrogen, and the traces were removed by keeping the samples under reduced pressure (about 13.3 Pa) for 30 min. Dried lipids were hydrated in buffer. Small unilamellar vesicles (SUVs) were formed by sonicating the samples above the main phase transition temperature of the lipids for at least 15 min. To determine the secondary structure, LTX-315 was added to the liposome suspension to obtain a peptide-to-lipid molar ratio of 15 mol%. The experiments were performed at 37 °C in an AquaSpec Cell (Bruker Optics, Ettlingen, Germany). The unit was integrated with a Tensor II spectrometer with a liquid nitrogen MCT using a spectral resolution of 4 cm^−1^ and 120 scans per spectrum. The secondary structure elements α-helices and β-sheets were predicted following the methods supplied by the Confocheck^TM^ system (Bruker Optics, Ettlingen, Germany). These methods were used to calculate the secondary structure using a multivariate partial least squares (PLS) algorithm based on a calibration dataset of 45 proteins.

## 5. Conclusions

Multicomponent lipid systems resembling the lipid composition of the cell membranes are crucial models to more adequately study the lipid behavior and the effect that some exogenous agents (such as peptides) can have on the membranes of interest through biophysical techniques. Our results showed that the electrostatic interaction between the cationic peptide LTX-315 and the negatively charged groups from tumoral cell membranes—such as phosphatidylserine—was involved in the molecular basis for the mechanism of action. This electrostatic attraction induced a destabilization of the lipid core of the tumoral model membrane that could explain the oncolytic activity previously reported for LTX-315 by Camilio et al. Additionally, our experimental data suggest that LTX-315 does not present conformational changes that can be associated with its biological activity. However, further peptide structure–activity studies are required to fully understand the mechanism of action and the biophysical interaction using representative model membranes.

## Figures and Tables

**Figure 1 ijms-24-00581-f001:**
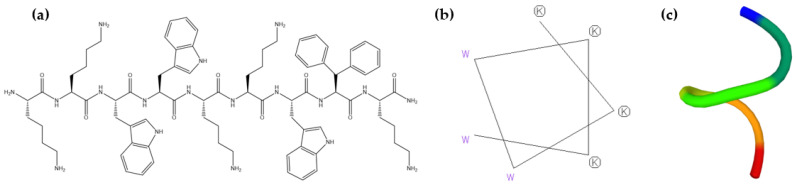
(**a**) Chemical structure, (**b**) helical wheel projection by EMBOSS, and (**c**) secondary structure prediction of LTX-315 by PEP-FOLD3. Each color represents an amino acid from the peptide sequence.

**Figure 2 ijms-24-00581-f002:**
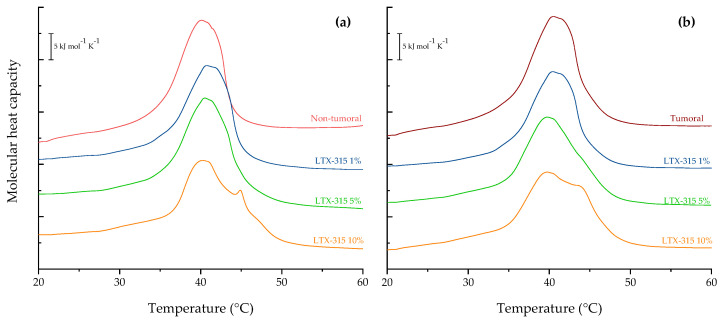
DSC heating thermograms of the (**a**) non-tumoral model membrane (DPPC/SM/DPPE 4.35:4.35:1 *w*/*w*) and (**b**) tumoral model membrane (DPPC/SM/DPPE/DPPS/4.35:4.35:1:0.3 *w*/*w*) in the presence of LTX-315 at different molar percentages: 1 mol% (**−**), 5 mol% (**−**), and 10 mol% (**−**).

**Figure 3 ijms-24-00581-f003:**
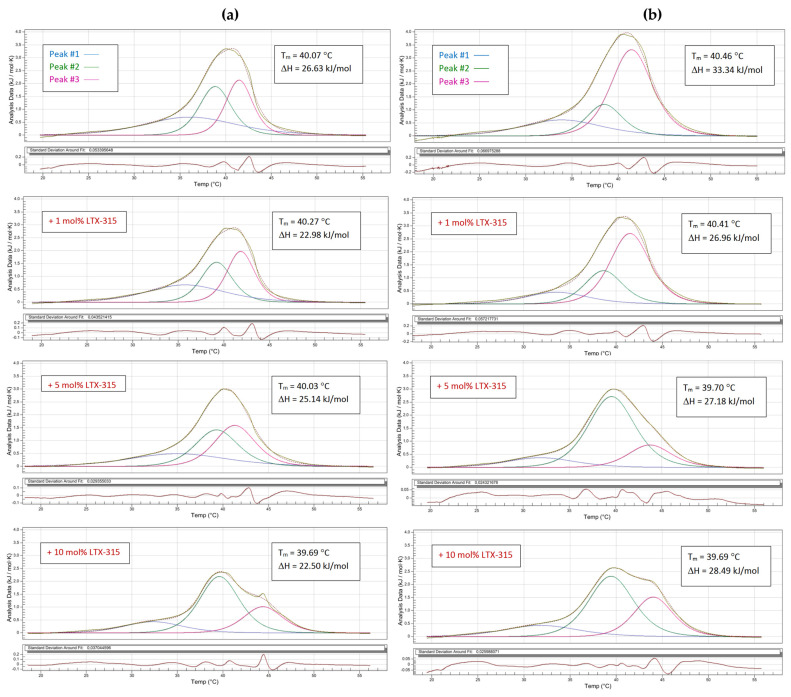
Deconvolution peaks of DSC heating curves obtained for MLVs prepared from the (**a**) non-tumoral model membrane (DPPC/SM/DPPE 4.35:4.35:1 *w*/*w*) and (**b**) tumoral model membrane (DPPC/SM/DPPE/DPPS/4.35:4.35:1:0.3 *w*/*w*) in the presence of LTX-315 at different molar percentages. Scans were obtained at a heating rate of 1 °C min^−1^.

**Figure 4 ijms-24-00581-f004:**
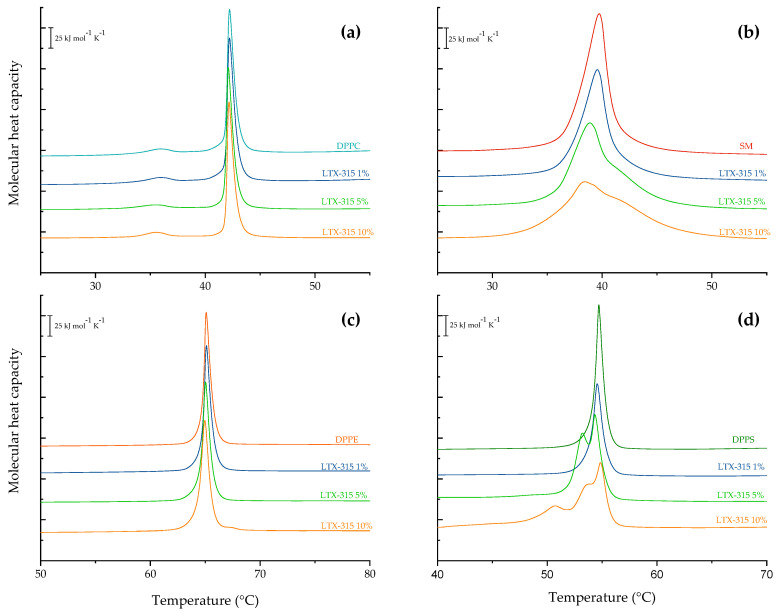
DSC heating thermograms of MLVs prepared from individual lipids—(**a**) DPPC, (**b**) SM, (**c**) DPPE, and (**d**) DPPS—at 1 mol% (**−**), 5 mol% (**−**), and 10 mol% (**−**) LTX-315.

**Figure 5 ijms-24-00581-f005:**
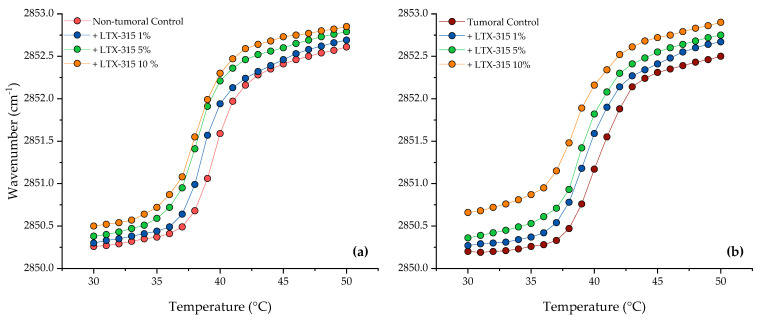
Peak position of the symmetric stretching vibration band of the methylene groups as a function of temperature: Effects of different LTX-315 concentrations at 1 mol% (-●-), 5 mol% (-●-), and 10 mol% (-●-) on (**a**) the non-tumoral model membrane (-●-) and (**b**) the tumoral model membrane (-●-).

**Table 1 ijms-24-00581-t001:** Thermodynamic parameters of deconvoluted peaks of representative heating endotherms registered for non-tumoral and tumoral vesicles with LTX-315: phase transition temperatures (T), full width at half-maximum (∆T_1/2_), peak height (Cp_max_), and transition enthalpy (∆H).

	Peak #1				Peak #2				Peak #3			
	T (°C)	∆T_1/2_ (°C)	Cp_max_ (kJ mol^−1^ K^−1^)	∆H (kJ mol^−1^)	T (°C)	∆T_1/2_ (°C)	Cp_max_ (kJ mol^−1^ K^−1^)	∆H (kJ mol^−1^)	T (°C)	∆T_1/2_ (°C)	Cp_max_ (kJ mol^−1^ K^−1^)	∆H (kJ mol^−1^)
Non-tumoral	36.20	14.69	0.69	9.48	38.86	3.98	1.88	8.02	41.48	3.37	2.13	8.39
+1 mol% LTX-315	35.84	11.22	0.69	7.69	39.20	3.57	1.80	7.10	41.84	3.16	2.24	8.07
+5 mol% LTX-315	35.14	18.56	0.76	7.24	39.32	3.88	2,02	8.79	41.32	4.79	1.84	9.39
+10 mol% LTX-315	32.35	9.38	0.43	3.92	39.61	4.69	2.45	12.87	44.40	5.20	1.01	5.85
Tumoral	33.98	8.26	0.87	7.20	38.51	3.57	1.47	19.78	41.46	5.00	3.31	6.26
+1 mol% LTX-315	33.62	8.77	0.45	4.15	38.68	4.90	1.28	15.95	41.53	5.10	2.71	7.10
+5 mol%LTX-315	31.95	11.22	0.38	3.72	39.58	6.32	2.71	18.40	43.71	6.12	0.87	5.36
+10 mol%LTX-315	31.87	11.42	0.38	4.81	39.46	5.71	2.33	14.97	44.03	4.90	1.51	8.32

**Table 2 ijms-24-00581-t002:** Phase transition temperatures (T_m_) of the synthetic lipid systems in the presence of several concentrations of LTX-315, as determined by FTIR. Standard deviations are ≤0.1 °C.

LTX-315 (mol%)	T_m_ (°C)	
	Non-Tumoral	Tumoral
0	39.7	40.4
1.0	38.4	39.7
5.0	38.3	39.4
10.0	38.3	38.8

## Data Availability

The data involved in this paper have been presented in articles and Appendix A in the form of diagrams or tables.

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
