# Peer review of "Understanding the Biophysical Interaction of LTX-315 with Tumoral Model Membranes"

_ijms, 2022, doi:10.3390/ijms24010581_

Round 1

Reviewer 1 Report

LTX-315 has been known as an important oncolytic peptide used in clinical trials against cancer. This work studies its interaction of with artificial tumoral and non-tumoral model membranes and found LTX-315 has a strong interaction with negatively charged phosphatidylserine.  

The main conclusion of this work is that the electrostatic interaction between the cationic peptide LTX-315 and the negatively charged groups from tumoral cell membranes, such as phosphatidylserine, is a significant molecular base for the mechanism of action. This is very apparent and not interesting enough for this field. 

Solidness: the experiment results are not strong enough to support the demonstration. For example:

Figure 5. How many samples are collected for this demonstration? What’s their statistic difference? If we look at Table 2, the Tm change in non-tumoral membrane is 1.4, which is very similar to 1.6 in tumoral membrane. The difference is negligible.

Research depth: not enough

Figure 2. The authors should dive deep to explain why the interaction between tumor molecular decreases the molecular heat capacity by referring existing findings or giving some educated hypothesis. 

Structure: 

Discussion section is too long and include lots of contents that should be put in instruction.

Format: 

Add legend in Figure 5. Change font of Figure 5 a and b to be consistent

Reviewer 2 Report

Klaiss-Luna et al. studied the interaction of LTX-315 with negatively charged tumoral model membranes through a series of biophysical studies. The results reported in this manuscript would highlight the mechanism of action of how these peptides interact with tumoral membranes. The narrative introduced in this manuscript is clear and the English usage is professional. The scientific methods and data analysis are sound. I therefore recommend this manuscript to be published Int. J. Mol. Sci. with some comments and suggestions below:

(1)   The authors should suggest, based on their observations, how to design and develop improved peptides with anticancer activity.

(2)   Are there any reported cytotoxicity data of LTX-315? It might be worth discussing the safety pattern of this peptide in the introduction.

(3)   Section 2.4 and figure 1c: It is recommended to perform a CD study to further support the predicted secondary structure.

(4)   Line 484 to 485: how do you make sure small unilamellar vesicles were formed with correct and uniform sizes? Maybe a DLS study would be needed to confirm that.

(5)   Please provide HPLC and MALDI traces in the SI.
